# High Pressure and Pasteurization Effects on Dairy Cream [note 1]

**DOI:** 10.3390/foods12193640

**Published:** 2023-10-01

**Authors:** Fernanda Machado, Ricardo V. Duarte, Carlos A. Pinto, Susana Casal, José A. Lopes-da-Silva, Jorge A. Saraiva

**Affiliations:** 1Associated Laboratory for Green Chemistry of the Network of Chemistry and Technology (LAQV-REQUIMTE), Department of Chemistry, University of Aveiro, 3810-193 Aveiro, Portugal; fernandamachado@ua.pt (F.M.); ricardo.vd@ua.pt (R.V.D.); carlospinto@ua.pt (C.A.P.); jals@ua.pt (J.A.L.-d.-S.); 2Associated Laboratory for Green Chemistry of the Network of Chemistry and Technology (LAQV-REQUIMTE), Laboratory of Bromatology and Hydrology, Faculty of Pharmacy, Porto University, Rua Jorge Viterbo Ferreira 228, 4050-313 Porto, Portugal; sucasal@ff.up.pt

**Keywords:** dairy cream, fatty acids, food safety, nonthermal and thermal processing, rheological parameters, shelf life, volatile organic compounds

## Abstract

Dairy cream, a common ingredient in various dishes and food products, is susceptible to rapid microbial growth due to its high water activity (≈0.97) and pH (≈6.7). Thus, it requires proper processing conditions to ensure food safety and extend shelf life. High-pressure processing (HPP) has emerged as a nonthermal food pasteurization method, offering an alternative to conventional heat-based techniques to obtain tastier, fresh-like, and safe dairy products without undesirable heat-induced alterations. This study assessed the impact of HPP (450 and 600 MPa for 5 and 15 min at 7 °C) and thermal pasteurization (75 °C for 15 s) on the microbiological and physicochemical attributes of dairy cream immediately after processing and throughout refrigerated storage (4 °C). HPP-treated samples remained microbiologically acceptable even on the 51st day of storage, unlike thermally pasteurized samples. Moreover, HPP decreased inoculated *Escherichia coli* and *Listeria innocua* counts by more than 6 log units to undetectable levels (1.00 log CFU/mL). pH, color (maximum variation of ΔE* up to 8.43), and fatty acid profiles remained relatively stable under varying processing conditions and during storage. However, viscosity exhibited higher values for HPP-treated samples (0.028 ± 0.003 Pa·s) compared to thermally processed ones (0.016 ± 0.002 Pa·s) by the 28th day of storage. Furthermore, volatile compounds (VOCs) of all treated samples presented a tendency to increase throughout storage, particularly acids and aliphatic hydrocarbons. These findings show HPP’s potential to significantly extend the shelf life of highly perishable dairy cream by at least 15 days compared to thermal pasteurization.

## 1. Introduction

Dairy cream is used as an ingredient in many products, including butter, ice cream, and sour cream, among others [1]. However, it is a highly perishable product, with a pH of around 6.7 and high water activity (around 0.97), requiring adequate preservation to increase its shelf life [2]. Traditionally, most cream for retail and industrial use is thermally pasteurized [3], aiming to destroy vegetative (pathogenic and spoilage) microorganisms and inactivate enzymes, extending the cream’s shelf life. Nonetheless, depending on the food matrix, heat pasteurization may not always be the ideal processing method since it may cause substantial modifications to the product’s optimal quality, including the development of off-flavors and the destruction of vitamins and other minerals. Consumers place a high value on food’s texture, flavor, aroma, shape, and color, and there is a growing demand for minimally processed, long-lasting products. As a result, alternative preservation techniques, particularly nonthermal ones, capable of preserving food’s sensory and nutritional qualities, have been tested and developed [4].

High-pressure processing (HPP) is a common nonthermal method that utilizes elevated hydrostatic pressures (approximately 400–600 MPa) to pasteurize, denature multiple enzymes, and inactivate pathogenic and spoilage vegetative microorganisms, thereby assuring food safety [5]. Unlike thermal pasteurization, HPP does not affect covalent bonds and is able to effectively retain food quality attributes, namely sensorial and nutritional properties [4]. Additionally, as a pasteurization technique, nonthermal HPP does not target bacterial spores, such as those from *Bacillus* spp., yet it can target some spores from yeasts and molds, with the exception of those from *Byssochlamys* and *Talaromyces* spp. and some species of *Zygosaccharomyces* [6]. As such, HPP products are to be kept under refrigeration.

Only a few studies have evaluated the effects of HPP on dairy creams [7,8]. One observed that 450 MPa treatment (10 or 25 °C during 15 or 30 min) followed by refrigerated storage (4 °C) for 8 days did not affect the fat globule size distribution and other physicochemical properties of pasteurized creams [8]. Regarding microbiological changes, another study showed that it is possible to considerably reduce *Listeria innocua* load in creams (35% fat), obtaining a decimal reduction time (D) of D450 MPa/25 °C for 7.4 min [9]. Differently, Gervilla et al. (2000) obtained D400 MPa/25 °C = 4 min on ewe’s milk (6% fat), showing the potential effect of fat to protect microorganisms against hydrostatic pressure [10].

Other methodologies have also been used for nonthermal pasteurization of dairy products, such as ultraviolet radiation, pulsed electric field (PEF), ultrasound, etc., yet these present lower efficacies compared to HPP, as PEF and US need to be combined with moderate temperatures to increase the inactivation rates of the target microorganisms, while ultraviolet radiation has low penetrance in opaque fluids. Other methodologies such as membrane filtration, despite its possible continuous use, require frequent cleaning and replacing the filters, which are rather expensive and do not allow a proper flow of bulky liquids through the filters [11].

To evaluate the impact and safety of this nonthermal technology and compare it with thermal pasteurization, (a) microbial load (endogenous and inoculated *Escherichia coli* and *L. innocua*), (b) fatty acid composition, (c) color parameters, (d) viscosity, and (e) volatile compounds were studied. Samples included the raw cream with no treatment (control), after the heat treatment (conventional pasteurization at 75 °C for 15 s), and after the pressure treatment (at 450 and 600 MPa for 5 min), followed by refrigerated storage (4 °C). The effect of HPP on inoculated microorganisms in dairy cream was also studied in a second set of experiments, at 600 MPa for 5 and 15 min.

## 2. Materials and Methods

### 2.1. Cream Samples

Industrially homogenized raw and thermally pasteurized (75 °C for 15 s) cream samples were kindly provided by a local cream-producing company (Portugal). Pasteurization was performed according to the commercial procedure used in the company [12].

### 2.2. Preparation of Cream Samples and Inoculation

Triplicated samples (20 mL each), for each storage day, were aseptically packed in UV-light sterilized low-permeability polyamide–polyethylene (PA/PE) bags and manually heat sealed prior to HPP, excluding as much air as possible.

Cultures of *E. coli* (ATCC 25922) and *L. innocua* (ATCC 33090) were grown in Tryptic Soy Broth (TSB; Liofilchem, Roseto degli Abruzzi, Italy) at 37 °C for 24 h to reach the stationary phase and then inoculated into raw cream to a final concentration of 108 cells/mL.

### 2.3. HPP Treatment of Samples

HPP treatments were performed in a pilot scale high-pressure device (Model 55, Hiperbaric, Burgos, Spain) with 55 L of vessel capacity, 2000 mm of vessel length, and 200 mm of vessel diameter. The pressure rise time was 200 MPa/min, and the decompression time was almost instantaneous. A first cream batch was subjected to 450 MPa and 600 MPa for 5 min each, at 7 °C, to optimize the pressure level required to achieve desirable microbial inactivation levels to extend the shelf life of dairy cream. Additionally, as described in the literature, the temperature of water increases between 2 and 3 °C for each 100 MPa [13]; so, in order to have a maximum temperature of 19–25 °C while at 600 MPa, the water temperature before pressurization was 7 °C. A second cream batch was processed at 600 MPa for 5 and 15 min at 7 °C to evaluate the effects of the processing time (at the most suitable pressure obtained in the first batch) on dairy cream. After the respective processing, samples from both batches were stored at 4 °C.

### 2.4. Storage Conditions

Thermally pasteurized and HPP samples from the first batch were stored under refrigeration (4 °C) for 5, 9, 18, 33, and 51 days, while samples from the second batch were stored for 3, 10, 28, and 52 days to evaluate and compare the shelf life of creams processed at both conditions (thermal pasteurization and HPP).

### 2.5. Microbial Analyses

After each experiment, cream samples from the first batch were analyzed for total aerobic psychrophiles (TAPs), Enterobacteriaceae (ENT), and lactic acid bacteria (LAB) counts. Apart from ENT, samples from the second batch were analyzed for the same microorganisms, along with inoculated *E. coli* (ATCC 25922) and *L. innocua* (ATCC 33090). Both cultures that were used to inoculate the cream samples were stored on Trypticase Soy Agar (TSA; Liofilchem, Roseto degli Abruzzi, Italy) Petri dishes at 4 °C. Briefly, one colony of each microorganism, previously isolated in TSA plate, was collected, inoculated in 250 mL of Tryptic Soy Broth (TSB; Liofilchem, Roseto degli Abruzzi, Italy), and incubated at 37 °C, 150 rpm, for 10–12 h. The growth period was selected in order to ensure that cells reached the stationary phase to be later inoculated into raw cream, with a final concentration of about 108 cells/mL. Under aseptic conditions, 20 mL of each cell suspension was used to inoculate 160 mL of the second batch of cream samples. The microbiological analyses were performed as described by [14]. The results were expressed as a decimal logarithm of colony-forming units per milliliter of cream (log CFU/mL). The maximum endogenous microbial load considered in this study was 6.00 log CFU/mL [15], and the detection limit was 1.00 log CFU/mL.

The experimental design of each cream batch is reported in Table 1.

### 2.6. pH and Color

The pH of all cream samples was measured at room temperature (21 ± 2 °C) in triplicate with a glass electrode (pH electrode 50 14, Crison Instruments, S.A., Barcelona, Spain).

The color was assessed using a Konica Minolta CM 2300d (Konica Minolta, Osaka, Japan) spectrophotometer on three random spots per sample, recorded according to the CIELab system, and the data were processed with the SpectraMagicTM NX software (Konica Minolta, Osaka, Japan). The obtained parameters were L*-lightness, a*-redness, and b*-yellowness. The total color difference (ΔE*) was calculated using Equation (1) [16].
ΔE* = [(L* – L*_0_)^2^ + (a* – a*_0_)^2^ + (b* – b*_0_)^2^]^1/2^(1)
where ΔE* is the total color change between a sample and the control (initial values identified with the subscript “0”).

### 2.7. Apparent Viscosity Measurements

The cream’s apparent viscosity was determined using a controlled-stress rheometer (AR-1000, TA Instruments, New Castle, DE, USA), equipped with a cone-and-plate geometry (acrylic cone, 6 cm diameter, and 2° angle). The bottom plate temperature was kept constant using a circulating bath (Circulating Bath 1156D, VWR International, Carnaxide, Portugal). Samples were equilibrated to 25 °C for about 15 min and then gently homogenized and placed carefully (approximately 2 mL) on the top of the bottom plate to minimize the damage to the sample structure and avoid trapping air bubbles. Flow curves were obtained by applying a continuous shear stress ramp (0 to 3 Pa) for 3 min [17].

### 2.8. Fatty Acid Determination

Fatty acids (FAs) were determined by gas chromatography as methyl esters (FAMEs). Briefly, fat was separated by centrifugation at 13,000 rpm for 20 min. Then, 40 µL of the upper layer (fat phase) was dissolved in hexane (2 mL), and the FAs were converted to their respective FAME by cold transmethylation (ISO 12966-2, 2011). Chromatographic separation was achieved with an Agilent J&W Select FAME column (100 m × 0.25 mm, J&W Agilent, Santa Clara, CA, USA) using a Chrompack CP 9001 gas chromatograph (Chrompack, Middelburg, The Netherlands) equipped with an FID detector. FA identification and FID calibration were accomplished with a certified reference standard mixture (TraceCert–Supelco 37 component FAME mix) and individual FAME, all from Supelco. Fatty acids were expressed in a relative percentage of their FAME.

### 2.9. Volatile Profile

The volatile compound (VOC) profiles were determined by headspace solid-phase microextraction (HS-SPME) followed by gas chromatography–mass spectrometry (GC-MS), as performed by [18], with modifications. Initially, 5 mL of each sample was placed in 20 mL headspace vials, and then cyclohexanone was added as an internal standard along with 28% sodium chloride (*w*/*w*). The vials were heated at 60 °C for 20 min with constant stirring (250 rpm), and the SPME fiber (DVB/CAR/PDMS; 50/30 µm; Supelco Inc., Bellefonte, PA, USA) was exposed for 30 min (60 °C). Volatiles were thermally desorbed for 5 min in the injector port (splitless mode; 250 °C). Chromatographic separation was performed on a fused-silica DB-5 MS column (30 m ×0.25 mm I.D. × 0.25 μm film thickness) from J&W Agilent (Santa Clara, CA, USA), with a temperature program going from 40 to 235 °C and a total run time of 60 min. The MS transfer line and ion source were at 280 °C and 230 °C, respectively, and the MS quadrupole temperature was at 150 °C, with an electron ionization of 70 eV set in full scan mode (m/z 40 to 650 at 1.2 scan/s). Compounds were identified by comparing their respective mass spectra with a mass spectral database (NIST v14, nist.gov, accessed on 21 September 2023). Semi-quantification was achieved as internal standard equivalents basis and expressed in μg of internal standard equivalents per 100 mL of cream.

### 2.10. Statistical Analysis

The experiments were performed in triplicate, each analyzed in duplicate. Statistical analysis of the results was performed using a two-way analysis of variance (ANOVA) followed by a multiple comparison post hoc test and Tukey’s honest significant differences (HSD) test at a 5% level of significance.

## 3. Results and Discussion

### 3.1. Microbial Analysis

Regarding the first batch experiments, TAP, LAB, and ENT were quantified before (initial) and right after (day zero) thermal or HPP, and also on days 5, 9, 18, 33, and 51 under refrigeration (4 °C) (Figure 1).

As represented in Figure 1, TAP, LAB, and ENT counts of thermally pasteurized cream samples decreased to below the detection limit (<1.00 log CFU/mL), maintaining low counts until the ninth day of storage. From the 18th day onwards, microbial growth was observed for all microorganisms, except for ENT, which remained undetected. By the 51st day, TAP and LAB counts surpassed 6.00 log CFU/mL, and, therefore, no further analyses were performed for the thermally pasteurized cream.

Right after processing at 450 MPa for 5 min, TAP loads observed were similar to those of raw dairy cream, while for samples processed at 600 MPa for 5 min, a very small (not statistically significant) decrease was observed. For ENT, regardless of the treatment, its counts were reduced to below 1.00 log CFU/mL and kept constant throughout storage. These results are in agreement with Permanyer et al. (2010), who reported a similar barosensivity of ENT when human milk was pressurized at 400, 500, and 600 MPa for 5 min at 12 °C [19]. Evert-Arriagada et al. (2014), working with starter-free fresh cheeses, also observed that after HPP (500 MPa, 5 min, 16 °C), ENT was not able to recover during all the cold storage period (21 days) [20]. From the 18th day onwards, both TAP and LAB counts were above 6.00 log CFU/mL; thus, samples treated at 450 MPa for 5 min (450/5) were considered spoiled. Samples processed at 600 MPa for 5 min (600/5) resulted in a slower recovery of TAP and LAB under refrigeration in comparison to those processed at 450/5, with TAP counts only increasing (*p* ≤ 0.05) after the 51st day (Figure 1). This demonstrates the efficiency of HPP at 600 MPa to injure microorganisms, taking them additional time to recover and develop compared to the thermal pasteurization process.

To evaluate the influence of pressurization time, a second study was performed, and a new fresh cream batch was processed at 600 MPa for 15 min (600/15) instead of 5. Since in the first study, ENT exhibited high sensitivity to both high pressure and pasteurization treatments, the effect of 600/15 was evaluated only for TAP and LAB. In addition to endogenous microorganisms, the effect of HPP (600/5 and 600/15) on inoculated *L. innocua* and *E. coli* was also evaluated.

The 600/15 condition significantly reduced (*p* ≤ 0.05) TAP and LAB counts by about 1.4- and 1.8-fold, respectively, compared to the initial raw cream counts (Figure 2).

By the 28th day of storage, TAP counts on thermally pasteurized samples increased to values above 6.00 log CFU/mL, while those treated by HPP (600/15) presented counts of 4.53 ± 0.11 log CFU/mL, evidencing the efficacy of HPP in inhibiting long-term microbial development and extending shelf life. Regarding LAB, a significant increase (*p* ≤ 0.05) throughout storage was observed.

Concerning inoculated microorganisms, 600/5 and 600/15 treatments were able to significantly reduce (*p* ≤ 0.05) *E. coli* counts compared to the initial inoculated load (Figure 2). By the 10th day, *E. coli* counts on both 600 MPa treatments experienced a significant increase (*p* ≤ 0.05), surpassing 6.00 log CFU/mL for 600/5 samples. However, on the following days, *E. coli* counts on both 600 MPa treated samples presented values below 1.00 log CFU/mL, probably due to the fact that *E. coli* is not able to survive after long exposures to low temperatures, as suggested by Arias et al. (2001) [21]. Despite the similar outcomes by the end of storage time, longer exposure to HPP at the same pressure appeared to be more effective in delaying microbial growth over time, given the lower counts registered on days 3 and 10.

Previous works revealed that gram-positive bacteria are more resistant to HPP than gram-negative [22,23]. In our study, *L. innocua* loads increased in both 600 MPa-treated samples (*p* ≤ 0.05) from the 28th up to the 52nd day of storage, suggesting that cells may recover from the injuries caused by HPP and grow during cold storage [24]. A larger number of L. monocytogenes cells on milk samples, after 10 days of refrigerated storage, was also observed by Liepa et al. (2018) [25]. This is probably due to the higher-pressure resistance of gram-positive bacteria, namely regarding their metabolic repair mechanisms, in comparison with gram-negative *E. coli*.

Even though thermal pasteurization was able to reduce initial microbial loads and inhibit microbial growth on the first days of storage, it is possible to conclude that HPP has a more pronounced effect on slowing microbial growth rate over time, as evidenced by lower microbial counts on the final day of storage (52nd).

Since milk and dairy products follow very strict regulations worldwide, further research is also needed to accurately establish the safety of dairy cream processed by HPP, namely to overcome these regulatory issues. For instance, in the United States, pasteurization must inactivate *Mycobacterium tuberculosis* and *Coxiella burnettii* (which is more heat stable than the first one), while also resulting in a negative phosphatase reaction [26].

### 3.2. pH and Color

The initial pH of the cream used in the first and second studies was similar to the ones reported in the literature [27]: 6.74 ± 0.05 and 6.91 ± 0.14, respectively (Appendix A). Regarding the first batch, all treated samples presented similar values to raw cream (*p* > 0.05), with small variations throughout storage (Appendix A). Contrarily, the pH of HPP samples was higher (*p* ≤ 0.05) during the first 9 days of storage, decreasing thereafter, which was probably caused by the observed microbial growth and the organic acids produced from their metabolic activity [28]. On the second batch, no significant differences (*p* > 0.05) between treatment conditions at each storage day were detected (Appendix A).

Regarding color measurements, detailed results for L*, a*, and b* values are presented in Appendix A (first batch) and Appendix A (second batch). In general, the L* parameter on both studies remained stable at all different storage days and conditions, except on 600/5 samples, where a significant increase (*p* ≤ 0.05) was observed when comparing the value obtained immediately after processing with that obtained on the 51st day of storage. For the a* parameter, it suffered some variations concerning both batches. On the first batch, compared to the initial raw cream, it was significantly higher (*p* ≤ 0.05) for both HPP samples and similar (*p* > 0.05) to the thermally pasteurized samples. On the contrary, in the second study, initial a* values of all samples, treated and non-treated, were statistically different (*p* ≤ 0.05) from each other, in the order from the highest a* value to the lowest: raw cream >600/15 > thermally pasteurized. These variations between the first and second studies are probably due to differences between the cream’s batch. By looking at every storage period, 450/5 and 600/5 samples remained statistically similar to each other (*p* > 0.05), differing only from thermally treated samples (*p* ≤ 0.05). The same happened with 600/15 and thermally treated samples on the second batch (*p* ≤ 0.05); the a* parameter on HPP samples was always higher than the thermally treated samples. Despite the small variations obtained for L*, a*, and b* parameters, no statistical differences (*p* > 0.05) were observed for the total color change (ΔE*) for all treatment conditions at each day of storage on both studies.

### 3.3. Viscosity

Cream’s flow behavior was studied only for raw, thermally pasteurized, and 600 MPa-processed samples in the second batch. Generally, samples showed a qualitatively similar non-Newtonian flow behavior, with apparent viscosity decreasing with shear rate (shear thinning). The observed behavior was expected for an emulsion and is in accordance with Donsì et al. (2011), who evaluated the rheological behavior of milk cream under pressure (400–500 MPa for 5–10 min at 25 °C) and reported that HPP milk cream also behaved as a non-Newtonian pseudoplastic fluid [29].

The apparent viscosity of the studied samples was compared at a constant shear rate of 33 s^−1^ (Table 2).

No major differences in the initial viscosity values were detected between the raw cream and all treated samples. Also, the apparent viscosity of the thermally pasteurized samples remained similar (*p* > 0.05) throughout the storage time. After 3 days of refrigerated storage, HPP samples presented viscosity values almost two times higher (*p* ≤ 0.05) than the initial ones (immediately after processing). However, from this day forward, viscosity values of HPP samples remained statistically similar (*p* > 0.05), only increasing (*p* ≤ 0.05) on the 52nd day of storage for 600/15 samples. Dumay et al. (1996) reported that after HPP, the flow behavior of the pasteurized cream samples did not show considerable changes after 7 days of storage (4 °C) [8]. In general, pressure-treated samples always presented a higher viscosity (*p* ≤ 0.05) than the heat treated ones (Table 2), which can be advantageous in the development of products with different texture characteristics and consumer acceptability.

### 3.4. Fatty Acid Analysis

To our knowledge, this is the first study that reports the changes observed in cream’s fatty acid profile after thermal pasteurization and pressurization treatments, upon storage. The GC analysis revealed the presence of twenty-nine FAs (Appendix A). The cream samples were essentially rich in saturated, followed by monounsaturated, and a small percentage of polyunsaturated FAs. Raw cream’s main fatty acids were palmitic (C16:0, 24.50 ± 0.12%), oleic (C18:1c, 20.03 ± 0.11%), and myristic (C14:0, 11.25 ± 0.07%), similar to what was previously reported [28,30].

In general, the different treatments did not affect (*p* > 0.05) saturated and monounsaturated FAs, while polyunsaturated decreased (*p* ≤ 0.05) on the 52nd day of storage. Regarding the main FA on cream, only C16:0 was present in higher (*p* ≤ 0.05) amounts in raw cream (compared to processed samples). Moltó-puigmartí et al. (2011) reported that HPP did not significantly change FA proportions compared to untreated human milk [31].

### 3.5. Volatile Analysis

A total of 39 different VOCs were identified in cream samples. Table 3 shows the chemical families of the VOCs and the total volatile amounts (identified and non-identified) detected. In general, there was a tendency for total volatiles to increase throughout storage. In raw cream, the most abundant families were aliphatic hydrocarbons, followed by aldehydes/ketones, acids, and lactones. Immediately after thermal and HPP treatments, a new class of compounds, alcohols, was detected on cream samples (Table 3). Alcohols can be produced by the reduction in their corresponding aldehydes and methyl ketones, through the activity of LAB dehydrogenases or by sugar fermentation, which is in accordance with the lower pH measured in these samples [32].

From the 28th day onwards, thermally treated samples presented higher (*p* ≤ 0.05) amounts of alcohol compared to HPP samples. Similarly, Chugh et al. (2014) studied the effect of thermal pasteurization on skim milk’s volatile composition, observing that during refrigerated storage, alcohol concentration increased as a result of the reduction in the corresponding carbonyl compounds [33].

The initial amount of acid compounds in raw cream increased (*p* ≤ 0.05) immediately after thermal and HPP treatments (Table 3). Throughout the storage, the number of acids on HPP samples remained higher (*p* ≤ 0.05) than on thermally treated samples. Garrido et al. (2015) observed a relevant increase of carboxylic acids in human milk after processing at 400 or 600 MPa for 6 min, which was probably due to the release of short-chain FAs from triglycerides (lipolysis) [34]. Acids can act as precursor molecules for a series of catabolic reactions, which can lead to the production of other flavor compounds such as alcohols, lactones, and methyl ketones [32].

All cream samples presented a similar (*p* > 0.05) content of aldehydes/ketones after processing compared to the raw cream, except for 600/15 samples, which presented higher levels (*p* ≤ 0.05) (Table 3). Vazquez-Landaverde et al. (2006) observed that at 25 °C, ketone concentration in milk processed under 620 MPa at 1, 3, or 5 min was similar to raw milk [35]. Despite the fact that ketones are naturally present in raw milk, most of them can be formed during heat treatment by β-oxidation of saturated fatty acids or by decarboxylation of β-ketoacids. Furthermore, several authors reported that HPP enhances the oxidation of free FAs, leading to the formation of ketone VOCs [34,36,37]. Vazquez-Landaverde et al. (2006) also observed an increase in aldehyde concentration when milk was processed at 620 MPa, which was possibly due to a higher solubility of oxygen under high pressure, which could enhance the formation of hydroperoxides, resulting in more aldehydes [35].

Aliphatic hydrocarbons were the major VOCs found on cream samples, presenting no regular tendency throughout the storage days under all processing conditions; they were statistically similar (*p* > 0.05) on both thermal and HPP samples. However, their content was significantly higher on thermally treated samples (*p* ≤ 0.05). Accordingly, Chugh et al. (2014) observed a significant increase in hydrocarbon compounds after heat treatment of skim milk [33].

Lactones, detected in very low levels in all cream samples, are related to lipid degradation and are formed by the cyclization of γ- and β-hydroxy acids [32]. Lactone levels in HPP samples were similar (*p* > 0.05) to raw cream but were higher in thermally pasteurized samples (*p* ≤ 0.05). Throughout the storage, lactones increased significantly on all treated samples (*p* ≤ 0.05) and were always higher on HPP samples.

In summary, initially treated samples were similar to raw cream with a general tendency to increase volatile amounts throughout the storage period, without major differences between heat pasteurized and HPP samples.

## 4. Conclusions

The present study evaluated the feasibility of using HPP for the nonthermal pasteurization of raw dairy cream as an alternative to the conventional heat-based pasteurization processes. HPP samples were still microbiologically acceptable by the 51st day of refrigerated storage, unlike thermally processed ones, which clearly highlights the use of this nonthermal technology to extend the shelf life of dairy cream. For the effect of HPP on inoculated microorganisms, even though HPP at 600 MPa for 15 min was able to reduce microbial loads to lower counts than 600 MPa for 5 min (at the beginning of the storage experiments), a similar microbiological development pattern was observed on both processing conditions by the end of the shelf life evaluation period, indicating that the inactivation effect is less likely to be dependent on processing time. In general, pH, color (maximum variation of ΔE* up to 8.43), and fatty acids (mainly palmitic, oleic, and myristic acids) were not considerably changed by the different processing conditions and storage, while viscosity presented higher values (*p* ≤ 0.05) for HPP samples (0.034 Pa·s, at the 52nd day). Furthermore, VOCs of all treated samples presented a tendency to increase throughout storage, particularly acids and aliphatic hydrocarbons. From a practical point of view, commercial (heat pasteurized) refrigerated dairy cream usually presents a shelf life <3 weeks. This shelf life could be considerably extended by at least 30 days using HPP, without major changes in the products’ quality, clearly evidencing the potential of this nonthermal technology for dairy cream pasteurization. Indeed, these results open the possibility of using HPP for the nonthermal pasteurization of dairy products, such as fresh cheeses, dairy creams, and even milk, either for retailing or using as food ingredients, as the extended shelf life can not only increase food safety but also reduce food waste.

## Figures and Tables

**Figure 1 foods-12-03640-f001:**
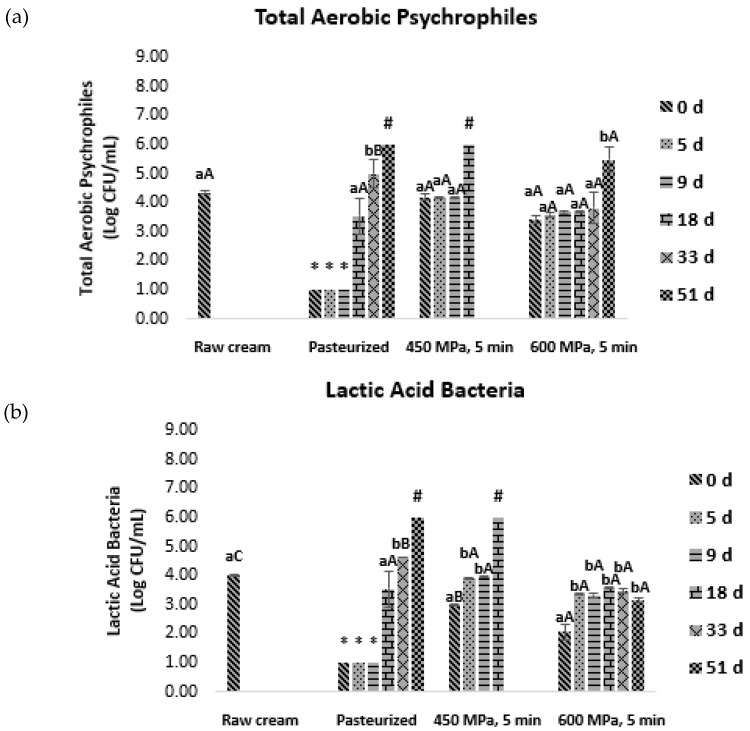
Microbial growth of (**a**) TAP, (**b**) LAB, and (**c**) ENT on initial raw cream after heat (75 °C, 15 s) treatment and after pressure treatment under 450 MPa and 600 MPa during 5 min (first batch). Analyses were made on the initial cream and right after processing (
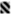
) and after 5 (
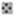
), 9 (
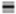
), 18 (
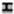
), 33 (
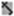
), and 51 (
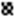
) days of storage at 4 °C. Bars with * and # represent microbial loads below the detection limit (lower than 1.00 log CFU/mL) and above 6.00 log CFU/mL, respectively. Different letters denote significant differences (*p* ≤ 0.05) between storage days for each condition (A,B) and treatment conditions for each storage day (a,b).

**Figure 2 foods-12-03640-f002:**
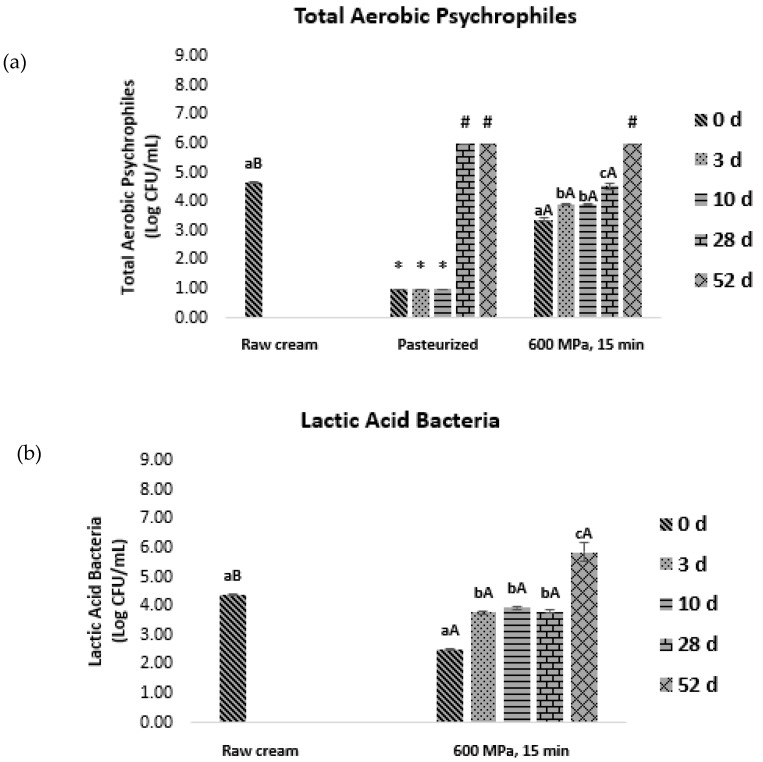
Microbial growth of (**a**) TAP, (**b**) LAB, (**c**) *E. coli*, and (**d**) *L. innocua* on initial raw cream after heat treatment (75 °C, 15 s) and after pressure treatment under 600 MPa for 5 min and 600 MPa during 15 min (second batch). Analyses were made on the initial cream and right after processing (
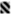
) and after 3 (
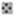
), 10 (
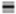
), 28 (
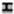
), and 52 (
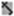
) days of storage at 4 °C. Bars with * and # represent microbial loads below the detection limit (lower than 1.00 log CFU/mL) and above 6.00 log CFU/mL, respectively. Different letters denote significant differences (*p* ≤ 0.05) between storage days for each condition (A,B) and treatment conditions for each storage day (a–c).

**Table 1 foods-12-03640-t001:** Experimental design of each cream batch and the aim of the study.

	HPP Conditions		Storage Period (Days)	Nomenclature	To Study the Effect of HPP after Processing and during Each Storage Period On:
	Pressure (MPa)	Duration (min)			
First batch	-	-	-	Raw	General microbiology (TAP, LAB, and ENT) and physicochemical parameters
	-	-	0, 5, 9, 18, 33, 51	Pasteurized	
	450	5		450/5	
	600	5		600/5	
Second batch	-	-	-	Raw	General microbiology (TAP and LAB) and physicochemical parameters Inoculated *E. coli* and *L. innocua*
	-	-	0, 3, 10, 28, 52	Pasteurized	
	600	5		600/5	
	600	15		600/15	

**Table 2 foods-12-03640-t002:** Apparent viscosity values determined at a particular shear rate (33 s^−1^) for initial raw cream and cream at different treatment conditions (75 °C, 15 s, 600/5, and 600/15) right after processing and after 3, 10, 28, and 52 days of storage at 4 °C. Results are presented as mean ± standard deviation. Different superscript letters denote statistical differences (*p* ≤ 0.05) between storage days for each condition (A–C) and treatment conditions for each storage day (a,b).

Storage Time (Days)	Conditions	Shear Rate (1/s)	Viscosity (Pa·s)
0	Initial	33.19	0.015 ± 0.001 ^aA^
	Heat treated		0.017 ± 0.001 ^aA^
	600 MPa/5 min		0.016 ± 0.001 ^aA^
	600 MPa/15 min		0.015 ± 0.001 ^aA^
3	Heat treated		0.018 ± 0.002 ^aA^
	600 MPa/5 min		0.031 ± 0.003 ^bB^
	600 MPa/15 min		0.026 ± 0.002 ^bB^
10	Heat treated		0.017 ± 0.002 ^aA^
	600 MPa/5 min		0.027 ± 0.003 ^bB^
	600 MPa/15 min		0.026 ± 0.002 ^bB^
28	Heat treated		0.016 ± 0.002 ^aA^
	600 MPa/5 min		0.028 ± 0.003 ^bB^
	600 MPa/15 min		0.030 ± 0.003 ^bBC^
52	Heat treated		–
	600 MPa/5 min		0.030 ± 0.003 ^aB^
	600 MPa/15 min		0.034 ± 0.003 ^aC^

**Table 3 foods-12-03640-t003:** Cream volatile profile (mg/100 g equivalents of cyclohexanone) at different treatment conditions (75 °C, 15 sec, 600/5, and 600/15) of the initial cream and right after processing (0 d) and after 3 (3 d), 10 (10 d), 28 (28 d), and 52 (52 d) days of storage at 4 °C. Results are presented as mean ± standard deviation. Different superscript letters denote statistical differences (*p* ≤ 0.05) between storage days for each condition (A–C) and treatment conditions for each storage day (a–d).

Storage Time (Days)	Conditions	Alcohols	Acids	Aldehydes/Ketones	Aliphatic Hydrocarbons	Lactones	Total Volatiles
0	Initial	Nd	16.4 ± 2.5 ^aA^	19.5 ± 1.3 ^aA^	51.6 ± 1.9 ^aAB^	0.2 ± 0.1 ^aA^	315.1 ± 26.9 ^aAB^
	Heat treated	3.2 ± 0.4 ^aAB^	33.5 ± 4.7 ^aAB^	36.9 ± 1.0 ^aAB^	80.5 ± 0.8 ^aB^	1.1 ± 0.1 ^aB^	492.8 ± 36.7 ^aB^
	600 MPa/5 min	4.1 ± 0.6 ^aAB^	54.8 ± 3.3 ^aB^	26.2 ± 4.6 ^aAB^	41.6 ± 0.4 ^aA^	0.4 ± 0.1 ^aAB^	291.1 ± 11.4 ^aA^
	600 MPa/15 min	6.6 ± 0.6 ^aB^	52.6 ± 6.4 ^aB^	47.5 ± 1.0 ^bB^	6.9 ± 0.2 ^aA^	0.4 ± 0.1 ^aAB^	314.1 ± 10.7 ^aAB^
3	Heat treated	3.6 ± 0.3 ^aA^	34.3 ± 2.8 ^aA^	49.6 ± 10.2 ^abB^	94.4 ± 2.8 ^abA^	0.7 ± 0.1 ^aA^	427.2 ± 64.5 ^aA^
	600 MPa/5 min	5.2 ± 0.3 ^aA^	114.2 ± 13.7 ^cB^	42.0 ± 0.7 ^abAB^	152.5 ± 3.8 ^bB^	0.9 ± 0.1 ^aAB^	493.9 ± 21.7 ^bA^
	600 MPa/15 min	4.4 ± 0.7 ^aA^	53.1 ± 0.7 ^aA^	23.0 ± 0.8 ^aA^	79.4 ± 2.9 ^bA^	1.6 ± 0.1 ^bB^	479.9 ± 58.2 ^aA^
10	Heat treated	4.1 ± 0.2 ^aA^	26.1 ± 3.7 ^aA^	60.5 ± 1.7 ^bAB^	121.6 ± 3.2 ^bA^	1.5 ± 0.3 ^aA^	373.5 ± 35.8 ^aA^
	600 MPa/5 min	6.6 ± 0.3 ^aA^	84.4 ± 4.8 ^bB^	70.8 ± 6.4 ^bB^	220.2 ± 13.4 ^cB^	2.4 ± 0.3 ^bB^	703.2 ± 66.1 ^cB^
	600 MPa/15 min	8.3 ± 0.2 ^aA^	78.9 ± 4.12 ^bB^	43.4 ± 8.0 ^abA^	310.2 ± 6.8 ^dC^	2.3 ± 0.2 ^bAB^	782.4 ± 32.4 ^bB^
28	Heat treated	20.3 ± 5.0 ^bB^	62.8 ± 9.8 ^bA^	74.5 ± 11.8 ^bA^	69.2 ± 5.7 ^aA^	2.6 ± 0.2 ^bA^	687.9 ± 37.8 ^bA^
	600 MPa/5 min	5.9 ± 1.2 ^aA^	152.5 ± 3.8 ^dB^	73.9 ± 2.6 ^bA^	346.5 ± 36.4 ^dB^	4.9 ± 0.3 ^cB^	1007.3 ± 49.9 ^dC^
	600 MPa/15 min	4.6 ± 0.4 ^aA^	220.7 ± 6.7 ^cC^	71.4 ± 6.4 ^cA^	83.3 ± 2.2 ^bA^	5.6 ± 0.1 ^cB^	797.6 ± 28.6 ^bB^
52	Heat treated	50.5 ± 3.7 ^cB^	126.1 ± 8.9 ^cA^	61.9 ± 12.7 ^bB^	64.2 ± 4.2 ^aA^	2.6 ± 0.2 ^bA^	751.2 ± 56.4 ^bA^
	600 MPa/5 min	4.1 ± 0.3 ^aA^	167.1 ± 6.4 ^dB^	52.2 ± 2.2 ^bAB^	119.9 ± 8.1 ^bB^	5.3 ± 0.5 ^cB^	783.3 ± 30.2 ^cA^
	600 MPa/15 min	6.1 ± 0.3 ^aA^	307.0 ± 3.3 ^dC^	36.2 ± 3.4 ^abA^	224.2 ± 1.2 ^cC^	5.7 ± 0.2 ^cB^	881.1 ± 112.7 ^bA^

Nd—not detected.

## Data Availability

Data are available upon request to the corresponding author.

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
