# Peer review of "High Pressure and Pasteurization Effects on Dairy Cream†"

_foods, 2023, doi:10.3390/foods12193640_

Round 1

Reviewer 1 Report

The manuscript entitled "High pressure and pasteurization effects on dairy cream" has interesting topic; however, i found two issues which should be addressed by the authors; 1. The similarity is 54% which is unacceptable for a journal like Foods, 2. i found many research/review papers published in this subject since 1996 as i googled it in the google scholar. The authors should explain how different their study is from others and what are the benefits of their study. 

Moderate improvements are needed.

Author Response

Referee #1

The manuscript entitled "High pressure and pasteurization effects on dairy cream" has interesting topic; however, i found two issues which should be addressed by the authors;

  1. The similarity is 54% which is unacceptable for a journal like Foods,

The authors acknowledge the issue raised by the referee. The similarity found between this paper and other documents is related to the fact that this paper results from the first authors’ master thesis (Fernanda Machado), which the university made freely available for consultation after two years after the presentation (which took place in 2019), which is available here: https://ria.ua.pt/bitstream/10773/27490/1/Documento.pdf. Additionally, a small part of this work was presented in a conference XXVI Encontro Nacional da Sociedade Portuguesa de Química, which took place in Porto (Portugal) (available in https://www.researchgate.net/publication/335002010_High_pressure_processing_and_thermal_pasteurization_effects_on_dairy_cream), being the reason why a similarity of 54% was found by the referee.

  1. I found many research/review papers published in this subject since 1996 as i googled it in the google scholar. The authors should explain how different their study is from others and what are the benefits of their study. 

The authors partially agree with the reviewer. The study from 1996 focused mainly from the physical standpoint of fat globule size distribution and flow behaviour of HPP-treated dairy cream and a model system consisting of a water-oil emulsion, and used processing conditions that are not suitable for industrial application. On the other hand, the present study investigates the impact of commercial high-pressure processing (HPP) conditions on dairy cream. We demonstrate that HPP-treated dairy cream exhibits extended microbiological stability, remaining acceptable even after an extended storage period, whereas thermally processed samples deteriorate earlier. It also shows that HPP maintains stable pH, color, and fatty acid profiles, preserving the quality of dairy cream. The volatile compounds analysis gives information on flavour and aroma changes, and their tendencies throughout storage.

Reviewer 2 Report

1.       I appreciate authors for nicely presented abstract.

2.       Give information on impact of HPP on yeast/ mould or spore forming bacillus species/spores.

3.       How commonly they are associated with cream.

4.       Line 71: it will be nice to mention local cream producing company instead of local industrial company.

5.       Provide GPS location for sample collated from.

6.       Provide clarity that HPP processing has been done for cream that already dispensed in final packing or final packing was the secondary after HPP.

7.       Section 2.5 should be written clearly for readers understanding. It is now confusing.

8.       Figure 1 and 2a: provide the units for 450/5 and 600/5……

9.       It will be nice to provide legends of graph in graph itself. This will make graphs self-explanatory.

10.   If only results were given for section 3, avoid discussion of results in results section. However, it seems that authors need to revise the section 3 title as results and discussion.

Need improvements 

Author Response

Referee #2

  1. I appreciate authors for nicely presented abstract.

The authors acknowledge the positive feedback.

  1. Give information on impact of HPP on yeast/ mould or spore forming bacillus species/spores.

When it comes to yeasts and molds, in a vegetative state, HPP can inactivate these structures at pressures above 400/450 MPa for a few minutes. When it comes to yeast and mold spores, some resistance may be found, specially for Zygosaccharomyces spp., Byssochlamys and Talaromyces spp., which require a combination of both high pressures and moderate/high temperatures in order to be inactivated. When it comes to bacterial spores, these are not inactivated by nonthermal HPP, being also required the combination of hydrostatic pressures and high temperatures for a successful inactivation. Despite this, the germination and outgrowth of spores can be considerably delayed by refrigeration, as both thermal and HPP are pasteurization methods, whose products will ultimately require cold storage to hurdle spores’ development.

This information was added to the manuscript, in the introduction section.

  1. How commonly they are associated with cream.

The most common sources of contamination in dairy products (in this case dairy cream) can arise from the animal itself (for example, the contact between the skin and the milk of the animal, equipment used for milking and the air itself on the farms. The most prevalent bacterial spores in dairy cream are those from Bacillus spp., and, in a lower extension, the fungi spores more prevalent are those from Byssochlamys spp. Yet, as long as the pasteurized products are kept under refrigeration conditions, the development of these structures is considerably delayed. This information was added to the manuscripts’ introduction.

  1. Line 71: it will be nice to mention local cream producing company instead of local industrial company.

Changed accordingly.

  1. Provide GPS location for sample collated from.

It is located in the north region of Portugal, nevertheless the company insists not to be identified for confidentiality issues.

  1. Provide clarity that HPP processing has been done for cream that already dispensed in final packing or final packing was the secondary after HPP.

The section 2.2 was changed to the following: “…were aseptically packed in UV-light sterilized low permeability polyamide-polyethylene (PA/PE) bags and manually heat-sealed, prior to HPP…”

  1. Section 2.5 should be written clearly for readers understanding. It is now confusing.

New information was added to Table 1 in order to clarify what is described in section 2.5.

  1. Figure 1 and 2a: provide the units for 450/5 and 600/5……

Changed accordingly.

  1. It will be nice to provide legends of graph in graph itself. This will make graphs self-explanatory.

Changed accordingly.

  1. If only results were given for section 3, avoid discussion of results in results section. However, it seems that authors need to revise the section 3 title as results and discussion.

Changed accordingly.

Reviewer 3 Report

The MS required revision see my comments in the pdf.

Author Response

Referee #3

Page 1:

  1. Lack in important results in the abstract

Changed accordingly. Please regard to the abstract section with the highlighted section.

  1. What are conditions of this treatment?

Changed accordingly.

  1. This is too much time

In this work we used longer storage time in order to understand microbial growth kinetics and the effectiveness of HPP in inhibiting such growth. Truly, when it comes to shelf-life evaluation, it is of most pertinence, especially from an economic perspective, to make it as long as possible in accurately infer the maximum period of time wherein the product is safe for consumption and does not loose quality. This was the main reason why the shelf-life assessment was so long.

  1. How many the shelf life by using this treatment?

The additional shelf-life obtained by using HPP was 15 days compared to thermal pasteurization.

  1. Add results of color components

Added accodingly

  1. This is too much time.

Please regard to the previous answer considering the selection of the sampling times for the shelf-life evaluation.

  1. Keywords must be arranged by alphpatical

Changed accordingly.

  1. This is no new technique.

Changed accordingly.

Page 2:

  1. What about the other non-thermal pasteurization method proberly better than hp. You must write about these methods.

Changed accordingly. Please check the introduction section containing this information

  1. Use one term "traditional or thermal"

Changed accordingly.

  1. Why you chosen this conditions?

These were the standard conditions selected by the company for pasteurisation of cream.

  1. Why the interval unequal?

As the authors were unaware of the time that microbial growth was going to occur, we started the analyses in short period of time, as as the results were being obtained, it was realized that the timeframe between sampling could be extended, as little to no microbial development was occurring in  the first sampling times wherein the analyses took place.

Page 3:

  1. Why you used thes conditions?

Both batches were necessary to perform two sets of experiments, in order to test different HPP conditions (time/pressure) and its effect on different microorganisms. The authors intended, with the first batch, to optimize the pressure level required to achieve desirable microbial inactivation levels to extend the shelf-life of dairy cream, while with the second batch the authors intended to evaluate the effects of the processing time (at the most suitable pressure obtained in the first batch) on dairy cream quality characteristics.

  1. This equation doe not belong to this reference. Use another fit reference to this equation.

Changed accordingly.

Page 7:

  1. Prsented these bacteria in pasteurized cream is danger.

The pathogenic-surrogated microorganisms, E. coli and L. innocua, were inoculated on dairy cream only with the purpose of evaluating the effects of HPP on these microorganisms in the case cream is highly contaminated. Furthermore, as discussed in the article, initially, HPP (600/15) successfully decreased these microorganisms counts (initial load approximately 8 log CFU/mL) on dairy cream to below detection limit (<1.00 log CFU/mL). This was indeed made to create the worst-case scenario of a heavily contaminated product, in order to achieve at least 5-log units’ reduction of pathogenic microorganisms to prove the pasteurization status.

Page 11:

  1. The results in this table unclear increasing, decreasing......?????

The authors agree with the referee. Truly, displaying the results in tables does not allow for a quick, visual perception of increasing/decreasing trends. Yet, considering the length of the manuscript (already with 15 pages), additional figures would made the manuscript even longer. So, the authors decided to compact the results and display them in tables.

Page 12:

  1. Support with important results.

Added accordingly.

Page 13:

  1. Add future prospective

Added accordingly.

Round 2

Reviewer 1 Report

To whom it may concern

The authors have made sufficient improvements to continue considering the publishing,.

Thank you

Reviewer 2 Report

Authors revised the manuscript extensively, thus I recommend to accept the said version.